# The Harmonious Interplay of Amino Acid and Monocarboxylate Transporters Induces the Robustness of Cancer Cells

**DOI:** 10.3390/metabo11010027

**Published:** 2021-01-02

**Authors:** Go J. Yoshida

**Affiliations:** Department of Immunological Diagnosis, Juntendo University Graduate School of Medicine, 2-1-1, Hongo, Bunkyo-ku, Tokyo 113-8421, Japan; go-yoshida@juntendo.ac.jp

**Keywords:** ASCT2 (SLC1A5), cancer stem-like cells, CD44 variant, glutamine addiction, LAT1 (SLC7A5), metabolic symbiosis, monocarboxylate transporter, redox stress, sulfasalazine, system Xc(-)

## Abstract

There is a growing body of evidence that metabolic reprogramming contributes to the acquisition and maintenance of robustness associated with malignancy. The fine regulation of expression levels of amino acid and monocarboxylate transporters enables cancer cells to exhibit the metabolic reprogramming that is responsible for therapeutic resistance. Amino acid transporters characterized by xCT (SLC7A11), ASCT2 (SLC1A5), and LAT1 (SLC7A5) function in the uptake and export of amino acids such as cystine and glutamine, thereby regulating glutathione synthesis, autophagy, and glutaminolysis. CD44 variant, a cancer stem-like cell marker, stabilizes the xCT antiporter at the cellular membrane, and tumor cells positive for xCT and/or ASCT2 are susceptible to sulfasalazine, a system Xc(-) inhibitor. Inhibiting the interaction between LAT1 and CD98 heavy chain prevents activation of the mammalian target of rapamycin (mTOR) complex 1 by glutamine and leucine. mTOR signaling regulated by LAT1 is a sensor of dynamic alterations in the nutrient tumor microenvironment. LAT1 is overexpressed in various malignancies and positively correlated with poor clinical outcome. Metabolic reprogramming of glutamine occurs often in cancer cells and manifests as ASCT2-mediated glutamine addiction. Monocarboxylate transporters (MCTs) mediate metabolic symbiosis, by which lactate in cancer cells under hypoxia is exported through MCT4 and imported by MCT1 in less hypoxic regions, where it is used as an oxidative metabolite. Differential expression patterns of transporters cause functional intratumoral heterogeneity leading to the therapeutic resistance. Therefore, metabolic reprogramming based on these transporters may be a promising therapeutic target. This review highlights the pathological function and therapeutic targets of transporters including xCT, ASCT2, LAT1, and MCT.

## 1. Introduction

Metabolic reprogramming specific to cancer cells is one of the ten cancer hallmarks described by Drs. Hanahan and Weinberg in their review article published in 2011 [1]. Some of the most striking alterations of tumor cellular bioenergetics include activation of glycolysis, increase in glutaminolytic flux, upregulation of amino acid transporters and lipid metabolism, enhancement of mitochondrial biogenesis, and activation of the pentose phosphate pathway and macromolecule biosynthesis [2,3].

Increasing evidence strongly suggests that metabolic reprogramming is crucial for cancer stem-like cells to maintain unlimited self-renewal potential and hyper-adaptation to drastic changes in the tumor microenvironment [4,5]. Cancer stem cells have a robust phenotype, encompassing several characteristics such as a slow cell cycle, the ability to detoxify or promote the efflux of anti-cancer drugs, resistance to redox stress, and a rapid response to genotoxic damage—all of which contribute to the acquisition of chemoresistance [6,7,8,9]. For example, while non-cancer stem-like cells are susceptible to chemotherapy and undergo apoptosis, released cytokine prostaglandin E_2_ (PGE_2_) awakens the dormant cancer stem-like cells localized in the niche, which is the favorable microenvironment [10,11]. Proliferating cancer stem-like cells are likely to exhibit additional metabolic reprogramming, concomitant with the upregulation of mitochondrial oxidative phosphorylation (OXPHOS)-related molecules. 

Due to the rapid proliferation, cancer cells have increased demand for amino acids in maintaining one-carbon metabolism, signal pathway, as well as the synthesis of nucleotide and protein [12,13]. The expression levels of amino acid transporters are closely associated with tumor size, pathological grade and distant metastasis [14,15,16]. Thus, increasing investigations have demonstrated the feasibility of amino acid transporters as a component of anti-cancer therapy.

This review highlights the pathological significance of amino acid transporters including xCT, ASCT2, and LAT1, as well as monocarboxylate transporters (MCTs) such as MCT1 and MCT4. Given that cancer cells exhibit the aerobic glycolysis termed “Warburg effect”, metabolites derived from glycolysis are important materials for amino acid production and macromolecule synthesis, which is required for robust tumor growth and proliferation. 

## 2. xCT (SLC7A11)

System Xc(-) is composed of a light-chain subunit (xCT, also known as SLC7A11) and a heavy-chain subunit (CD98hc, also referred to as SLC3A2), and functions as a Na^+^-independent transporter that mediates the exchange of extracellular cystine for intracellular glutamate [17,18]. The availability of cysteine is a rate-limiting factor for the synthesis of the reduced form of glutathione (GSH), and the activity of system Xc(-) is crucial for the GSH-dependent anti-oxidant machinery. xCT interacts with the type II transmembrane protein CD98hc at the cell surface. CD98hc is separately and covalently linked to several light chains that function as amino acid transporters at the plasma membrane, including LAT1, LAT2, y+LAT1, y+LAT2, ASC-1, and xCT [19]. Mice lacking xCT expression appear healthy, but they have an increased plasma concentration of cystine compared with their wild-type littermates [20], suggesting that xCT-mediated cystine transport is required for cells exposed to severe redox stress. Transcription of the *xCT* gene is induced by depletion of cystine or redox stress due to electrophilic agents, and this effect is mediated by binding of the transcription factor Nrf2 to its response element in the promoter of the *xCT* gene [21]. Exposure of normal airway epithelial cells to cigarette smoking upregulates xCT [22] through the transient activation of the Nrf2 signaling pathway [23]. In many cases of non-small cell lung cancer (NSCLC), constitutive activation of Nrf2 in the nucleus caused by the loss-of-function *KEAP1* genetic mutation prevents redox stress accumulation triggered by chemotherapy [24]. This is consistent with the poor 5-year overall survival of xCT-overexpressing NSCLC patients [22]. System Xc(-) is a main regulator of metabolic reprogramming with overarching effects on glucose metabolism, glutamine dependency, and intracellular GSH/glutathione disulfide redox balance in cancer stem-like cells. Furthermore, activating transcription factor 4, which plays an essential role in the response of cells to multiple types of stress [25], upregulates xCT expression [26,27]. These observations indicate that xCT contributes to the protection of cancer cells exposed to a high level of oxidative stress.

Stabilization of xCT at the plasma membrane of cancer cells is regulated by CD44v8-10 (an isoform that includes the sequence encoded by variant exons 8−10) and promotes GSH synthesis [6,28]. Small-interfering RNA-mediated depletion of CD44v8-10 downregulates xCT expression at the cellular membrane, thereby exhausting the intracellular cysteine pool without affecting the intracellular content of other amino acids. Therefore, CD44v8-10 plays a fundamental role in the GSH-dependent antioxidant system in cancer cells by modulating xCT-mediated cystine transport and consequently GSH synthesis. Alternative splicing of CD44 mRNA regulated by epithelial splicing regulatory protein 1 (ESRP1) induces CD44v8-10 expression in metastatic tumor-initiating cells [6,29]. Cancer cells positive for CD44v8-10 exhibit an enhanced capacity to defend against redox stress as a result of increased xCT-mediated cystine uptake and GSH synthesis. Such tumor-initiating cells with high levels of GSH are thus predominantly responsible for colonization of the pre-metastatic niche in the lungs, which contains neutrophils that generate oxidative stress [29,30].

A growing body of evidence suggests that the crosstalk between cancer cells and stromal cells plays an important role in metabolic reprogramming [4,31,32,33,34]. Metabolic interactions between chronic lymphocytic leukemia (CLL) cells and neighboring bone marrow-derived stromal cells (BMSCs) promote cancer cell survival. If cysteine is not provided by BMSCs, CLL cells, which lack the xCT transporter, are susceptible to redox stress. Neighboring stromal cells characterized by BMSCs can take up cystine via system Xc(-) and release cysteine [31]. A vast majority of cells, including tumor cells, do not depend on extracellular cysteine for GSH synthesis [31,35,36]. Instead, they take up cystine (two cysteine molecules joined by a disulfide bond), the more abundant and more stable oxidized form, and reduce it to cysteine in the cytoplasm [37]. By striking contrast, CLL cells aggressively import cysteine and use it for the synthesis of GSH, resulting in increased resistance to oxidative stress. Under these conditions, CLL cells can display enhanced therapeutic resistance to F-ara-A and oxaliplatin, two chemotherapeutic agents used clinically to treat CLL [31]. Blocking xCT activity in BMSCs with sulfasalazine (SSZ) suppresses CLL cell viability and improves the effectiveness of anticancer drugs.

SSZ, which is used for the treatment of rheumatoid arthritis and ulcerative colitis, inhibits system Xc(-), thereby targeting the CD44v8-10-xCT interaction [28,29,38]. SSZ, a well-characterized specific inhibitor of xCT-mediated cystine transport, suppresses CD44-driven tumor growth and activates p38 mitogen-activated protein kinase signaling (Figure 1). Ishimoto et al. showed that the combination of SSZ and cisplatin (CDDP) significantly decreases tumor proliferation compared with CDDP alone [28], suggesting that SSZ reduces the capacity of cancer stem-like cells to protect against redox stress and sensitizes them to available chemotherapeutic agents. In addition, SSZ inhibits the formation of metastatic lesions in the lungs derived from CD44v8-10-expressing breast cancer cells [29]. Alternative splicing of CD44 mRNA regulated by ESRP1 increases xCT-dependent resistance to redox stress, thereby allowing tumor cells to evade exogenous stress in the pre-metastatic niche. Cancer stem-like cells of head and neck squamous cell carcinoma (HNSCC) that survive treatment with cetuximab, an epidermal growth factor receptor (EGFR)-targeting agent, are sensitive to the induction of ferroptosis by SSZ [38,39]. CD44v8-10-expressing and undifferentiated cancer stem-like cells exhibit resistance to the monoclonal anti-EGFR antibody and sensitivity to SSZ [38]. This finding suggests that the Achilles’ heel of CD44v8-10-positive cancer stem-like cells may depend on the regulation of oxidative stress by xCT. Given the intratumoral heterogeneity in terms of the association between CD44v8-10 and xCT, combination therapy with SSZ and cetuximab may be an effective treatment.

## 3. ASCT2 (SLC1A5)

ASCT2, also known as SLC1A5, promotes the uptake of circulating glutamine into proliferating tumor cells [40]. ASCT2 is overexpressed in squamous cell carcinoma, adenocarcinoma, and neuroendocrine lung cancers [41]. In addition, overexpression of ASCT2 in oral squamous cell carcinoma is positively correlated with poor outcomes [42]. ASCT2 serves as an obligatory exchanger that imports a sodium-coupled amino acid substrate into cells and exports another sodium-coupled amino acid substrate with 1:1 stoichiometry [43,44]. ASCT2 is the primary transporter importing glutamine, and its inhibition attenuates tumor growth, which partially explains “glutamine addiction” [15,40,41]. Glutamine flux, which is dependent on the balance between the uptake of glutamine by ASCT2 and its subsequent export by LAT1 (SLC7A5), leads to high intracellular availability of essential amino acids (EAAs). Once in the cytoplasm, glutamine is a substrate of several glutamate-producing enzymes, such as mitochondrial glutaminase (GLS1), and cytosolic enzymes involved in the biosynthesis of nitrogenous metabolites. Glutamine-derived glutamate is likely to be transported back out of the cancer cell in exchange for cystine by system Xc(-) [45]. The combination of amino acid transporters plays a critical role in metabolic reprogramming and maintenance of the stem-like phenotype in cancer cells. For instance, in triple-negative breast cancer (TNBC) cells, the interaction between ASCT2 and xCT causes “glutamine addiction” [46,47]. System Xc(-) takes up cystine in exchange for glutamate for GSH synthesis, whereas ASCT2 imports glutamine in a cooperative manner [28,48] (Figure 2). Thus, targeted knockdown of ASCT2 inhibits GSH synthesis and induces ferroptosis mediated by the accumulation of intracellular redox stress [42].

Cancer cells cannot survive in the absence of exogenous glutamine, and therefore exhibit “glutamine addiction”, which is orchestrated by the interaction between xCT and ASCT2 [4,40,49]. Glutamine taken up through ASCT2 is rapidly exported via the bidirectional amino acid transporter LAT1 in exchange for the uptake of extracellular EAA. Knockdown of ASCT2 in cancer cells impairs glutamine uptake and export, uptake of EAAs, and mTORC1 activation, which suggests that both uptake and export of glutamine are required for the activation of mTORC1 signaling [40,50]. Mounting evidence strongly suggests that glutamine is an essential substrate required for anabolic growth in mammalian cells. Investigation with the transaminase inhibitor amino-oxyacetic acid indicates that the main route of entry of glutamine-derived carbon into the tricarboxylic acid cycle (TCA cycle) in Myc-transformed cells is through the transamination reaction [51]. Oncogenic c-Myc induces the transcription of glutamine transporters, such as ASCT2 and LAT1, and the expression of glutamine-utilizing enzymes, such as GLS1 [52,53]. Following the transport of glutamine into the cell, the first step of glutaminolysis is the GLS1-mediated conversion of glutamine into glutamate. Glutamate is subsequently converted to α-ketoglutarate (α-KG) by either glutamate dehydrogenase (GDH) or aminotransferases. Glutaminolysis consists of a series of biochemical reactions by which glutamine is catabolized into metabolites including α-KG and glutamate [54]. In the TCA cycle, α-KG is catabolized to malate, which is transported into the cytoplasm and converted to pyruvate and ultimately to lactate [55]. Mechanistically, mTORC1 signaling promotes glutamine anaplerosis mediated by upregulation of GDH [56]. In addition to promoting glutamine uptake, c-Myc facilitates the metabolism of imported glutamine into glutamic acid and ultimately into lactic acid, pyruvate, and aspartate [57]. On the other hand, l-γ-glutamyl-*p*-nitroanilide, one of the inhibitors of ASCT2, can block glutamine uptake and inhibit glutamine-dependent mTOR activation [58,59]. In the absence of amino acids, mTOR signaling is refractory to the stimulation of growth factors. The uptake of glutamine mediated by ASCT2, followed by its rapid efflux via LAT1 in exchange for EAAs such as leucine, is the rate-limiting step for mTOR signaling activation in tumor cells [59]. The interaction between the glutamine antiporter ASCT2 and the heterodimeric LAT1/CD98hc bidirectional transporter is necessary for amino acid transport and activation of the mTORC1 signaling pathway leading to cellular growth and autophagy. The generation of α-KG from the catabolism of glutamine is essential for the activation of mTOR signaling in cervical carcinoma and osteosarcoma cells [60]. Targeting glutamine uptake and glutaminolysis in cancer patients could inhibit mTOR signaling, even in the presence of aberrant growth factor stimulation. 

Induction of redox stress by treatment with SSZ requires ASCT2-mediated glutamine uptake and the synthesis of α-KG mediated by GDH in CD44v8-10-expressing cancer stem-like cells in HNSCC [61]. Increased expression levels of xCT and ASCT2 are positively correlated with the undifferentiated phenotype in HNSCC. The transcriptional program regulated by c-Myc is likely to play a critical role in glutaminolysis in CD44v8-10-positive undifferentiated HNSCC cells. Unlike CDDP, SSZ eliminates CD44v8-10-positive undifferentiated cancer cells, especially HNSCC cells that are also positive for ASCT2 [61]. Therefore, xCT-targeted therapeutic strategies may be effective for the depletion of ASCT2-positive cancer stem-like cells. Indeed, plasma membrane proteins, including CD44v, xCT, and ASCT2, are widely expressed in several types of human malignancy, such as HNSCC and colorectal cancer, and their upregulation is associated with poor prognosis [62,63]. 

## 4. LAT1 (SLC7A5)

The L-type amino acid transporter family is a crucial route of entry of EAAs into cancer cells and comprises four members (LAT1–4). Among the four LAT transporters, LAT1 is predominantly overexpressed in a variety of cancers [64,65,66,67]. LAT1 (SLC7A5) contains 12 transmembrane domains and covalently binds to CD98hc, which incorporates LAT1 into the cellular membrane, resulting in its functional expression [68]. LAT1 mediates the uptake of neutral EAAs (leucine, isoleucine, phenylalanine, methionine, histidine, tryptophan, valine, and tyrosine) into cancer cells [69,70] in exchange for the efflux of intracellular substrates (EAAs and/or glutamine) [71,72], thus serving as an amino acid antiporter. The complex composed of LAT1 and CD98hc functions as an antiporter that imports branched amino acids such as leucine and exports glutamine [50]. Nicklin et al. used pharmacological inhibitors and small-interfering RNAs to show that the inhibition of LAT1-CD98hc prevents mTORC1 activation by glutamine and leucine without affecting glutamine uptake [59]. Further, glutamine acts upstream of leucine as an efflux solute, allowing the uptake of leucine through the LAT1-CD98hc antiporter. 

LAT1 regulates the cellular uptake of exogenous leucine, thereby promoting mTORC1 activation in tumor cells [64,67,73]. Nutrient and growth factor pathways control cellular metabolism and protein synthesis by regulating mTORC1 activation [74,75]. Aberrant activation of mTORC1 is common in human malignancies [64,76], and mTORC1 signaling promotes tumor progression by stimulating signaling pathways that support cancer cell growth, autophagy, and resistance to apoptosis [75,77,78]. mTORC1 phosphorylates ULK1 under hypo-nutrient conditions, thereby preventing its activation by AMP-activated protein kinase (AMPK), a key activator of autophagy [76,79]. The relative activity of mTORC1 and AMPK in different cellular contexts largely determines the extent of autophagy induction. Among amino acids, leucine is the most effective activator of mTORC1 [74,76]. The expression level of LAT1 is regulated by activating transcription factor 4 (ATF4) depending on mTORC1 [80]. Lack of nutrition caused by amino acid deficiency activates ATF4, which induces amino-acid transport into cancer cells [80,81]. Increased ATF4 levels upregulate LAT1, causing an enhanced uptake of leucine and thereby activating mTORC1 and simultaneously inhibiting autophagy [80,82]. Indeed, the knockout of LAT1 or ATF4 can block amino acid uptake, prevent mTORC1 activation, and enhance autophagy.

The mTOR signaling pathway regulated by AMPK and LAT1 functions as a sensor of dynamic alterations in the nutrient tumor microenvironment [83,84]. DRAM1 binds amino acid transporters such as LAT1 and ASCT2, directing them to lysosomes and permitting efficient activation of mTORC1 [85]. The LAT1/ASCT2 and xCT/CD98hc complexes in cancer cells activate the mTORC1-SIRT4-GDH axis and anti-oxidant GSH synthesis, respectively [4,46,56]. The former pathway promotes the conversion of glutamate into α-KG, whereas the latter pathway regulates the level of redox stress. EpCAM, a functional marker of ovarian cancer stem cells, forms a complex with amino acid transporters including LAT1 and CD98hc [86,87,88], and the expression level of LAT1 is positively correlated with poor clinical outcomes in ovarian cancer, renal cell carcinoma, and pancreatic ductal adenocarcinoma [89,90,91]. LAT1 is markedly expressed in ovarian cancer, and positive LAT1 expression is an independent predictor of poor overall survival in patients with ovarian cancer [91]. By contrast, the expression of ASCT2 is not a significant prognostic factor of worse clinical outcomes in ovarian cancer, although positive expression of LAT1 is closely correlated with that of ASCT2 and CD98hc, and leads to significantly worse outcomes [91,92,93]. 

Increasing evidence suggests that LAT1 is involved in the metastatic potential of malignancies. LAT1 upregulation is positively correlated with metastasis in multiple cancers [82,89,94,95,96]. For example, lymph node metastasis-positive squamous cell carcinomas express LAT1, whereas a positive signal for LAT1 is not detected in metastasis-negative cells [95]. The transcriptional levels of LAT1 are significantly higher in renal cell carcinoma with metastasis [89]. Cells with high LAT1 expression tend to have more severe metastatic disease in gastric cancer [96]. LAT1 expression in neuroendocrine tumors is significantly associated with lymph node metastasis [94]. The functional significance of LAT1 in metastasis has been demonstrated. Knockdown of LAT1 inhibits the migration and invasion of gastric carcinoma and cholangiocarcinoma cells [97,98]. From the perspective of cancer treatment, heptane-2-carboxylic acid (BCH) inhibits proliferation and migration in human epithelial ovarian cancer cells without affecting Akt signaling [59,99]. These findings suggest that blockage of LAT1 is a promising strategy to prevent metastasis of cancer.

## 5. Monocarboxylate Transporters

A significant increase in the expression levels of MCT1 and MCT4 is a hallmark of several human malignancies, and high levels of these transporters lead to poor clinical outcome. For instance, increased MCT1 expression is detected in a wide range of malignancies, including glioma and neuroblastoma, as well as breast, colorectal, gastric, and cervical cancers [100,101,102,103,104,105]. MCT4 expression is highly elevated in renal cell carcinoma, as well as in cervical and prostate cancers [102,106,107]. Increased expression of MCT1 and MCT4 in cancer provides a therapeutic window for disabling these transporters with small-molecule inhibitors and/or by targeting their co-chaperone CD147 [108,109].

Post-translational stabilization of MCT1 has been observed under nutrient stress conditions [109]. It involves a poorly characterized mechanism associated with redox stress derived from OXPHOS in mitochondria [110]. Although glucose deprivation induces autophagy and activates canonical Wnt/β-catenin signaling, β-catenin downregulation decreases MCT1 expression in cancer cells, thereby positively coupling autophagy to high MCT1 expression [111]. In addition, MCT1, MCT4, and CD147 interact with the hyaluronate receptor CD44 in breast and prostate cancer cells [112,113]. In this complex, CD44 serves as a chaperone for MCT1 and MCT4, and impairment of CD44 signaling decreases the expression levels at the cellular membrane and impairs the activity of both MCT1 and MCT4 [113]. Depletion of the chaperone associated with MCT causes the inappropriate expression of MCT in intracellular vesicles, indicating that CD147 targets the transporters to the cellular membrane [114,115]. *CD147* gene expression is induced by hypoxia [116], which accounts for hypoxia-inducible MCT1 expression. Increased stability of MCT1 mRNA is related to the loss of the MCT1 translation repressor microRNA (miR)-124, as observed in pancreatic ductal adenocarcinoma and medulloblastoma [117,118]. Loss of miR-342-3p, which acts downstream of the estrogen receptor, upregulates MCT1 expression in TNBC cells [119]. MCT4 expression is indirectly regulated by miR-1, which is regulated by decreased levels of Smad3-hypoxia-inducible factor 1 (HIF-1) signaling, ultimately leading to the downregulation of MCT4 in glycolytic cancer cells [120]. Surprisingly enough, miR-1 inhibits Warburg effect while Smad3 promotes it, mediated by its downstream effecter HIF-1α and glycolytic enzymes such as hexokinase 2 (HK2) and MCT4 [120]. These findings strongly suggest that the transcriptional levels of MCT1 and MCT4 are both directly and indirectly regulated by miRNAs.

Metabolic interaction between epithelial tumor cells and cancer-associated fibroblasts (CAFs) requires that each cell population expresses different subtypes of MCT. Epithelial cancer cells express MCT1, whereas CAFs lacking caveolin expression are positive for MCT4 [121,122]. Cancer cells synthesize pyruvate from lactate, thereby providing the TCA cycle with an intermediate metabolite. An extracellular space rich in lactate leads to an acidic microenvironment, which in turn contributes to the generation of pseudo-hypoxic conditions. In this emerging concept of reverse Warburg effect, MCT1-positive cancer cells play an important role in maintaining the hierarchy in tumor cellular society unlike MCT4-positive CAFs [4,33,34]. In tumor cells positive for MCT1, a second signaling activity of lactate is tightly linked to its positive regulation of amino acid metabolism. In those cells, inhibition of HIF prolylhydoxylases by lactate-derived pyruvate leads to stabilization and activation of both HIF-1α and HIF-2α [109,123]. Interestingly, HIF-2α plays a fundamental role in the regulation of expression levels of ASCT2 in MCT1-positive cancer cells [123]. HIF-2α activates c-Myc signaling, thereby promoting glutamine uptake and metabolism through the upregulation of the inward glutamine transporter ASCT2 and the glutamine-metabolizing enzyme GLS1 [123]. 

However, all the types of cancer cells do not necessarily exhibit the reverse the Warburg effect. Tumors that express high levels of MCT4 or those with the mesenchymal phenotype fail to exhibit the reverse Warburg effect. Instead, hierarchical metabolic heterogeneity occurs: while MCT4-positive cancer cells depend on aerobic glycolysis and secrete lactate mediated by MCT4, MCT1-positive cancer cells import lactate through MCT1 and exhibit OXPHOS in mitochondria. In addition, the amount of glucose uptake is low in MCT1-positive cancer cells as compared with that in MCT4-expressing cells [124,125]. This metabolic heterogeneity is referred to as metabolic symbiosis, and this type of lactate shuttle is also observed between neurons and astrocytes in normal brain tissues [126]. It is notable that normal and cancerous tissues share finely regulated mechanisms of metabolic symbiosis. Remarkably, well-oxygenated cancer cells, which express high levels of MCT1, efficiently produce metabolic intermediates, as well as ATP and lactate, by utilizing lactate derived from hypoxic tumor cells expressing high levels of MCT4 (Figure 3). Redox stress is a major hallmark of malignant neoplasms, which drive the robust metabolism in adjacent proliferating MCT1-positive cancer cells in which a high amount of mitochondria exists; this is mediated by the paracrine transfer of mitochondrial fuels characterized by lactate, pyruvate, and ketone bodies [124,125].

Genotoxic stress due to chemotherapy and/or irradiation, which increases oxidative stress, promotes a stem-like phenotype [8,127,128,129]. As cancer stem-like cells exhibit a rapidly proliferating and poorly differentiated phenotype, MCT1-positive cancer cells are likely to harbor a stem-like phenotype in the heterogeneous cellular society. Indeed, cancer cells follow a hierarchical model in which a subpopulation of cancer stem-like cells have a tumorigenic potential much greater than that of other differentiated cancer cells [6,130]. The activated mitochondrial metabolism in cancer stem-like cells provides enough energy not only for self-renewal by symmetric cell division, but also for invasive and metastatic phenotypes. Therefore, pharmacological blockage of MCT1 is useful for the treatment of malignancy because the suppression of MCT1 activity inhibits metabolic symbiosis, and MCT1-positive aerobic cancer stem-like cells can no longer take up lactate [125]. These findings suggest that MCT1-positive cells play an important role in maintaining the hierarchy in cancer cellular society, in contrast to MCT4-positive cells.

## 6. Conclusions

Mounting evidence helps us understand the reason why cancer cells develop metabolic phenotypes that differ from those of adjacent, non-malignant tissues, as well as when these phenotypes represent actionable therapeutic vulnerabilities. Aberrant proliferation of tumor cells is maintained by the adaptation to a nutrient microenvironment generated through alterations in energy metabolism specific to malignancy. As a consequence, metabolic reprogramming is one of the hallmarks of cancer cells in parallel with genomic instability, chronic inflammation in the tumor microenvironment, and escape from the immune surveillance. Although aerobic glycolysis, also known as the Warburg effect, is a characteristic metabolic feature of cancer cells, recent investigations show that other metabolic features driven by the interaction of amino acid transporters, in particular, glutamine addiction, metabolic symbiosis, and reverse Warburg effect, are responsible for therapeutic resistance. Thus, metabolic reprogramming orchestrated by transporters such as xCT, ASCT2, LAT1, and the MCT family is a therapeutic target as the Achilles’ heel of malignant neoplasms.

## Figures and Tables

**Figure 1 metabolites-11-00027-f001:**
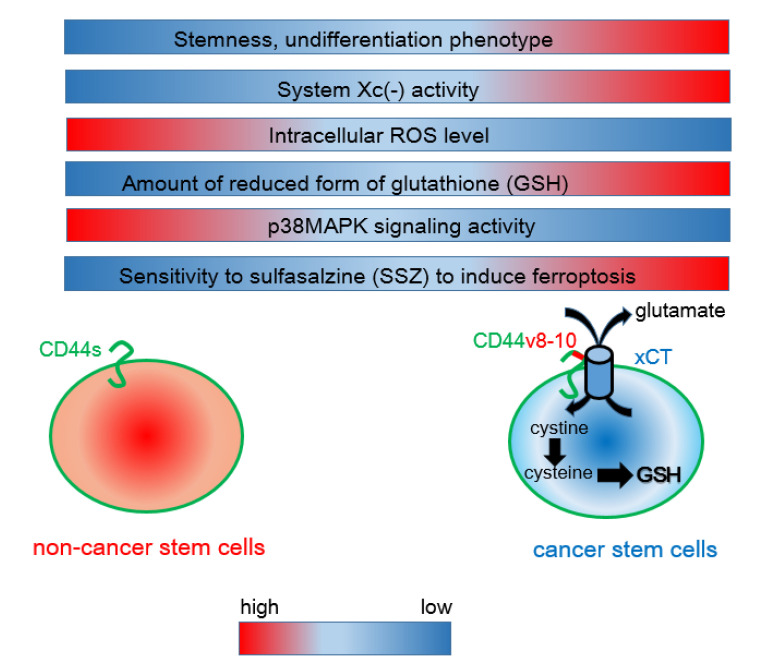
Interaction between CD44 variant 8-10 (CD44v8-10) and the xCT antiporter determines the resistance to oxidative stress. Unlike mesenchymal cancer cells expressing CD44 standard (CD44s), CD44v8-10-positive cancer stem-like cells exhibit robust system Xc(-) activity, resulting in enhanced uptake of cystine. Given that cysteine is a rate-limiting factor for the synthesis of the reduced form of glutathione (GSH), CD44v8-10-expressing cancer cells show a phenotype of increased resistance to redox stress. Sulfasalazine (SSZ) inhibits the xCT transporter and is therefore effective for blocking the CD44v8-10-xCT-GSH axis.

**Figure 2 metabolites-11-00027-f002:**
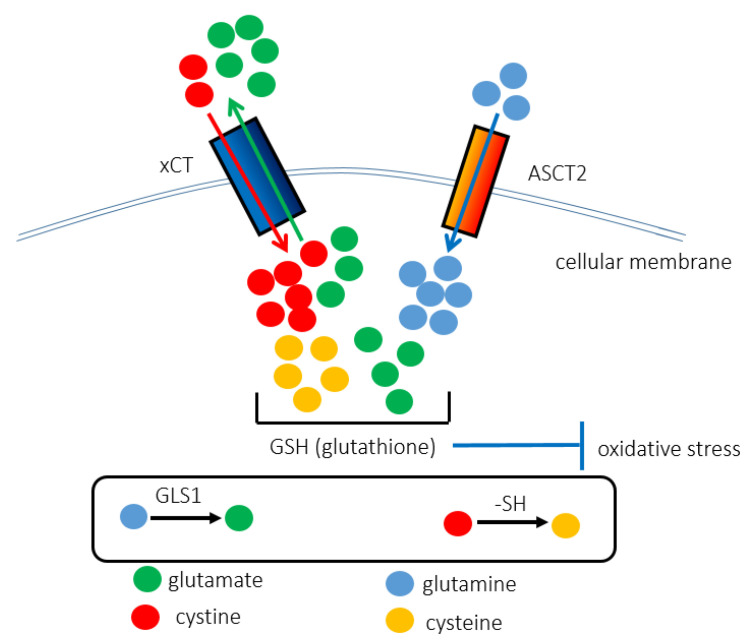
The interaction between ASCT2 and xCT contributes to the synthesis of GSH, rendering cancer cells resistant to redox stress. The xCT (SLC7A11) antiporter is responsible for importing cystine instead of exporting glutamate. ASCT2 (SLC1A5) promotes the uptake of glutamine. Glutamine and cystine are chemically converted into glutamate and cysteine, respectively. As GSH is composed of cysteine, glutamate, and glycine, the cooperative function of ASCT2 and xCT underlies the anti-oxidant stress machinery. That is why ASCT2-targeted therapy increases intracellular redox stress levels and induces ferroptosis, iron-dependent cell death.

**Figure 3 metabolites-11-00027-f003:**
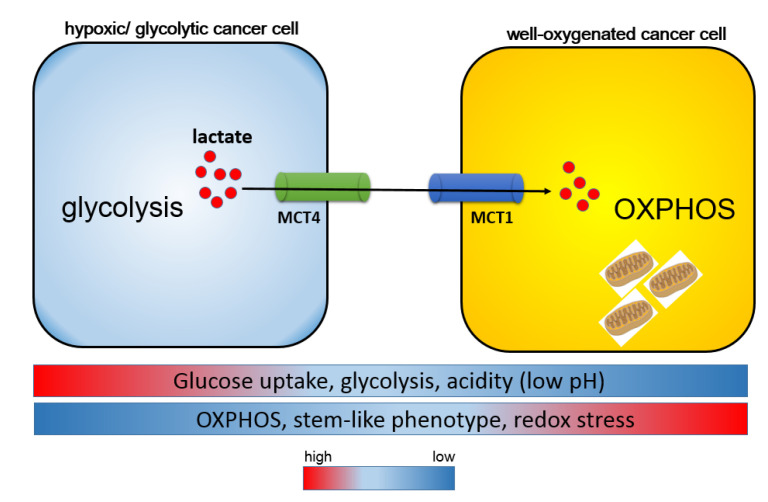
Harmonious interaction between MCT1 and MCT4 is responsible for the heterogeneity of cancer metabolism. Metabolic symbiosis occurs between well-oxygenated/aerobic cancer cells and hypoxic/glycolytic cancer cells. Intra-tumoral heterogeneity induces a lactate shuttle between hypoxic and oxidative cancer cells driven by both MCT1 and MCT4. In contrast to MCT1-expressing cancer cells, glucose uptake is robust in MCT4-positive cells. MCT4-positive hypoxic cells are responsible for the formation of an acidic tumor microenvironment through aerobic glycolysis and the secretion of lactate, whereas MCT1-positive oxidative cells utilize lactate as a metabolic intermediate for the tricarboxylic acid cycle (TCA cycle) and oxidative phosphorylation (OXPHOS), and consequently exhibit a stem-like phenotype.

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
