# Peer review of "The Harmonious Interplay of Amino Acid and Monocarboxylate Transporters Induces the Robustness of Cancer Cells"

_metabolites, 2021, doi:10.3390/metabo11010027_

Round 1
Reviewer 1 Report
This review manuscript summarizes the current efforts to understand how metabolic reprogramming regulated by specific transporters plays a role in cancer biology. In general, this manuscript is solid and provides important details, especially for the metabolic heterogeneity; the reviewer would suggest making this a separate section if possible. However, there are many areas overlapping to each other with repetitive sentence (e.g. mTOR signaling), especially for the crosstalk among these transporters. It is strongly suggested to re-structure the contents. For example, briefly discussing how and why targeting expression and stability of each transporters is important at various levels (transcription, translation, cofactors, stability, etc), followed by a separate section to cover their interactions since they appear exhibit a complex crosstalk. Other specific comments are listed below.
- One major issue in this review is the clarification of cancer stem cells vs cancer cells. Carrying variable molecular signatures, cancer stem cells and cancer cells respond to chemotherapies and environmental stress differently. Please make sure the statement in this review correctly captures such difference and the specificity, and also aligns with clear references to back it up.
- Recommend to clearly specify "amino acid and monocarboxylate transporters" in the title.
- ABSTRACT: The reviewer suggested to add a sentence to state the link between metabolic reprogramming and transporters in between the first two sentences in order to flow better.
- INTRODUCTION: The reviewer suggested to add a paragraph for introducing amino acid and monocarboxylate transporters before the last paragraph in INTRODUCTION section, such as the definition, the classification and the biological/pathological functions in general, before diving into the discussion of specific transporters.
- The colors in Figure 1 are confusing. Are the colors for two cell types correlated to the colors for the level of events?
- Page 4, third sentence: It would be helpful to specify what in vitro endpoints were used to determine "poor outcomes".
- Page 4: Add the abbreviation after glutaminase, GLS1.
- Figure 2: Suggested to add arrows between glutamine --> glutamate and cystine --> cysteine. The current visual looks like all four molecules are present independently. Although glutamine and cystine can be imported into the cytosol, glutamate and cysteine seem to be magically present.
- Page 5, last paragraph: Repetitive sentences to the next paragraph regarding mTORC1 signaling and ASCT2. It is suggested to either combine them together or remove this one.
- Page 6, last paragraph in ASCT2 section: Figure 2 is not a scheme for therapeutic strategies.
- Page 6, LAT1 section: The reviewer suggested to consolidate the entire LAT1 section. The last two sentences on page 6 can be moved to earlier section for importance of mTOR signaling in cancer.
- Page 7, last paragraph: What “the cell membrane expression” does the author refer to?
- Page 7, last sentence: Please clarify it should be CD147 or CD44.
- Page 8, last sentence in the first paragraph: It is recommended to state that “MCT1 and MCT4 are both directly and indirectly regulated by miRNAs”.
- Page 8, last sentence in the second paragraph: This sentence circles back to ASCT2 and GLS1 all of a sudden.
- Figure 3: The color for hypoxic/glycolytic cancer cells is confusing with the level of event.
Author Response
Reviewer #1
This review manuscript summarizes the current efforts to understand how metabolic reprogramming regulated by specific transporters plays a role in cancer biology. In general, this manuscript is solid and provides important details, especially for the metabolic heterogeneity; the reviewer would suggest making this a separate section if possible. However, there are many areas overlapping to each other with repetitive sentence (e.g. mTOR signaling), especially for the crosstalk among these transporters. It is strongly suggested to re-structure the contents. For example, briefly discussing how and why targeting expression and stability of each transporters is important at various levels (transcription, translation, cofactors, stability, etc), followed by a separate section to cover their interactions since they appear exhibit a complex crosstalk. Other specific comments are listed below.
One major issue in this review is the clarification of cancer stem cells vs cancer cells. Carrying variable molecular signatures, cancer stem cells and cancer cells respond to chemotherapies and environmental stress differently. Please make sure the statement in this review correctly captures such difference and the specificity, and also aligns with clear references to back it up.: The following sentence has been inserted to clearly describe the relationship between cancer stem cells and therapeutic resistance. “Cancer stem cells have the robust phenotype, encompassing several characteristics such as a slow cell cycle, the ability to detoxify or promote the efflux of anti-cancer drugs, resistance to redox stress, and a rapid response to genotoxic damage, all of which contribute to the acquisition of chemoresistance.”
This sentence is accompanied by the following references.
Yoshida, G. J.; Saya, H., Therapeutic strategies targeting cancer stem cells. Cancer Sci 2016, 107, (1), 5-11.
Makena, M. R.; Ranjan, A.; Thirumala, V.; Reddy, A. P., Cancer stem cells: Road to therapeutic resistance and strategies to overcome resistance. Biochim Biophys Acta Mol Basis Dis 2020, 1866, (4), 165339.
Recommend to clearly specify "amino acid and monocarboxylate transporters" in the title: Thank you for your productive comment. The title of the manuscript has been replaced by “The harmonious interplay of amino acid and monocarboxylate transporters induces the robustness of cancer cells.”
ABSTRACT: The reviewer suggested to add a sentence to state the link between metabolic reprogramming and transporters in between the first two sentences in order to flow better: Given the productive comment, I added the following sentence in the Abstract. “The fine regulation of expression levels of amino acid and monocarboxylate transporters enables cancer cells to exhibit the metabolic reprogramming leading to the therapeutic resistance.”
INTRODUCTION: The reviewer suggested to add a paragraph for introducing amino acid and monocarboxylate transporters before the last paragraph in INTRODUCTION section, such as the definition, the classification and the biological/pathological functions in general, before diving into the discussion of specific transporters. : Thank you for a precious comment, and I added the following paragraph; “Because of the rapid proliferation, cancer cells have increased demand for amino acids in maintaining one-carbon metabolism, signal pathway, as well as the synthesis of nucleotide and protein. The expression levels of amino acid transporters are closely associated with tumor size, pathological grade and distant metastasis. Thus, increasing investigations have demonstrated the feasibility of amino acid transporters as a component of anti-cancer therapy.”
The colors in Figure 1 are confusing. Are the colors for two cell types correlated to the colors for the level of events? : While red bar indicated high levels of ROS level and p38MAPK signaling activity in non-cancer stem cells with CD44 standard expression, blue bar showed high levels of system Xc(-) and sensitivity to SSZ in cancer stem cells expressing CD44 variant. That is why I strongly believe that the current format of the Figure 1 is suitable.
Page 4, third sentence: It would be helpful to specify what in vitro endpoints were used to determine "poor outcomes". : Reference [31] analyzed the patients with oral squamous cell carcinoma (OSCC), which is why the term of “poor outcomes” is considered to be suitable.
Page 4: Add the abbreviation after glutaminase, GLS1: Given the precious comment, I added “(GLS1)” just after mitochondrial glutaminase on page 4.
Figure 2: Suggested to add arrows between glutamine --> glutamate and cystine --> cysteine. The current visual looks like all four molecules are present independently. Although glutamine and cystine can be imported into the cytosol, glutamate and cysteine seem to be magically present.: Because additional arrows would make Figure 2 more complicated, I would like not to change the conventional version.
Page 5, last paragraph: Repetitive sentences to the next paragraph regarding mTORC1 signaling and ASCT2. It is suggested to either combine them together or remove this one. : Given the precious comment, I decided to combine the two paragraphs.
Page 6, last paragraph in ASCT2 section: Figure 2 is not a scheme for therapeutic strategies.: As you pointed out, Figure 2 is not directly relevant to the therapeutic resistance, I erased “(Figure2)” on page 6. Instead, I inserted “(Figure 2)” at the end of the sentences explaining the association between xCT and ASCT2.
Page 6, LAT1 section: The reviewer suggested to consolidate the entire LAT1 section. The last two sentences on page 6 can be moved to earlier section for importance of mTOR signaling in cancer. : Because LAT1 is closely related to autophagy, I added the following sentences to the paragraph; “Expression level of LAT1 is regulated by activating transcription factor 4 (ATF4) depending on mTORC1. Lack of nutrition caused by amino acid deficiency activates ATF4, which induces amino-acid transport into cancer cells. Increased ATF4 levels upregulate LAT1, causing an enhanced uptake of leucine and thereby activating mTORC1 and simultaneously inhibiting autophagy. Indeed, the knockout of LAT1 or ATF4 can block amino acid uptake, prevent mTORC1 activation, and enhance autophagy.”
Page 7, last paragraph: What “the cell membrane expression” does the author refer to?: As you kindly pointed out, this phrase is very confusing, so that it should be replaced by “decreases the expression levels at the cellular membrane.”
Page 7, last sentence: Please clarify it should be CD147 or CD44. : I clarify that CD147 is tightly associated with both MCT1 and MCT4 and facilitates their cell surface expression.
Page 8, last sentence in the first paragraph: It is recommended to state that “MCT1 and MCT4 are both directly and indirectly regulated by miRNAs”. : Thank you for providing me a meaningful and alternative sentence.
Page 8, last sentence in the second paragraph: This sentence circles back to ASCT2 and GLS1 all of a sudden. : I am appreciated for an important comment. Herein, I would like to mention the role of HIF2-alpha in the regulation of ASCT2, one of amino acid and transporters in MCT1-expressing cells. Thus, I added the following sentence; “Interestingly, HIF-2a plays a fundamental role in the regulation of expression levels of ASCT2 in MCT1-positive cancer cells.”
Figure 3: The color for hypoxic/glycolytic cancer cells is confusing with the level of event. : While hypoxic cancer cells activate glucose uptake and glycolysis, well-oxygenated cancer cells exhibit OXPHOS, oxidative stress, and stem-like phenotype. Furthermore, MCT1 and MCT4 should be pictured with the different colors. That is why I would like to maintain the original version of Figure 3.
Reviewer 2 Report
This is an interesting review outlining the role of transporters in relation to malignant and stem-like properties of cancer cells. The review is well written and my comments are minor suggestions.
Page 2, last line. Please provide a reference for this statement.
Fig. 2: Please depict ASCT2 as an antiporter, similar to xCT.
Page 4, last line: did you mean glutamine addiction?
Page 5, first line: in exchange for glutamate
Page 5, line 3: "Thus, targeted knockdown of ASCT2 inhibits
GSH synthesis and induces apoptosis mediated by the accumulation of intracellular redox stress." This statement is very general and not supported by systematic knock-out experiments (see Bröer S. Amino Acid Transporters as Targets for Cancer Therapy: Why, Where, When, and How. Int J Mol Sci. 2020 Aug 26;21(17):6156.)
Page 6, line 11: A major product of glutaminolysis is aspartate, which should be mentioned here.
Page 6, line 11: L-γ-glutamyl-p-nitroanilide is a very non-specific glutamine analogue with multiple actions (see reference above).
Page 6, line 15: Please note that the Nicklin model does not work in all cancer cells see (Bröer A, Rahimi F, Bröer S. Deletion of Amino Acid Transporter ASCT2 (SLC1A5) Reveals an Essential Role for Transporters SNAT1 (SLC38A1) and SNAT2 (SLC38A2) to Sustain Glutaminolysis in Cancer Cells. J Biol Chem. 2016 Jun 17;291(25):13194-205.)
Page 6, line 16: "The high affinity between the glutamine antiporter ASCT2 and the heterodimeric LAT1/CD98hc bidirectional transporter is necessary for amino acid transport and activation of the mTORC1" This statement suggests a direct complex between ASCT2 and LAT1, for which there is little evidence.
Page 7, line 8: It seems to be an odd mechanism for a cancer cell to upregulate aerobic glycolysis and to upregulate miR-1 to inhibit aerobic glycolysis. Is there more insight into this conundrum?
Author Response
Reviewer #2
This is an interesting review outlining the role of transporters in relation to malignant and stem-like properties of cancer cells. The review is well written and my comments are minor suggestions.
Page 2, last line. Please provide a reference for this statement. : Thank you for an important suggestion. I inserted the following references.
Bonifacio, V. D. B.; Pereira, S. A.; Serpa, J.; Vicente, J. B., Cysteine metabolic circuitries: druggable targets in cancer. Br J Cancer 2020.
Bansal, A.; Simon, M. C., Glutathione metabolism in cancer progression and treatment resistance. J Cell Biol 2018, 217, (7), 2291-2298.
Fig. 2: Please depict ASCT2 as an antiporter, similar to xCT. : For the description of ASCT2 as an antiporter, I added the following sentence; “ASCT2 serves as an obligatory exchanger which imports a sodium-coupled amino acid substrate into cells and exports another sodium-coupled amino acid substrate with 1:1 stoichiometry.”
Page 4, last line: did you mean glutamine addiction?: “addition to glutamine metabolism” has been corrected by “glutamine addiction.”
Page 5, first line: in exchange for glutamate: As you pointed out, “glutamine” has been corrected by “glutamate.”
Page 5, line 3: "Thus, targeted knockdown of ASCT2 inhibits GSH synthesis and induces apoptosis mediated by the accumulation of intracellular redox stress." This statement is very general and not supported by systematic knock-out experiments (see Bröer S. Amino Acid Transporters as Targets for Cancer Therapy: Why, Where, When, and How. Int J Mol Sci. 2020 Aug 26;21(17):6156.) : Systemic knock-out experiments are expected to be unnecessary to mention in this review manuscript, because the outcome of the specific depletion of ASCT2 in cancer cells should be investigated. In addition, I have replaced “apoptosis” by “ferroptosis.” Furthermore, I cited the above review paper by Professor Stefan Bröer in the section of Introduction.
Page 6, line 11: A major product of glutaminolysis is aspartate, which should be mentioned here. : In addition to lactic acid, I have clearly written the “pyruvate and aspartate” as the product of glutaminolysis.
Page 6, line 11: L-γ-glutamyl-p-nitroanilide is a very non-specific glutamine analogue with multiple actions (see reference above). : That is why I mention this agent as “one of the inhibitors of ASCT2.”
Page 6, line 15: Please note that the Nicklin model does not work in all cancer cells see (Bröer A, Rahimi F, Bröer S. Deletion of Amino Acid Transporter ASCT2 (SLC1A5) Reveals an Essential Role for Transporters SNAT1 (SLC38A1) and SNAT2 (SLC38A2) to Sustain Glutaminolysis in Cancer Cells. J Biol Chem. 2016 Jun 17;291(25):13194-205.) : Thank you for an important suggestion. It is sure that Nicklin et al. by using HeLa cells demonstrated that L-glutamine uptake is regulated by ASCT2 (SLC1A5) and loss of SLC1A5 function inhibits cell growth and activates autophagy. However, I think it is not necessary to clearly mention the type of cancer which Nicklin et al. mainly used for this research. After all, HeLa cells show the generality to assess the function of metabolism associated with malignancy.
Page 6, line 16: "The high affinity between the glutamine antiporter ASCT2 and the heterodimeric LAT1/CD98hc bidirectional transporter is necessary for amino acid transport and activation of the mTORC1" This statement suggests a direct complex between ASCT2 and LAT1, for which there is little evidence. : Given that relative to other neutral amino acid transporters, the expression levels of ASCT2 and LAT1, are coordinately elevated in a wide spectrum of primary human cancers, suggesting that they are frequently co-opted to support the "tumor metabolome." That is why I decided to replace “the high affinity” by “the interaction.”
Page 7, line 8: It seems to be an odd mechanism for a cancer cell to upregulate aerobic glycolysis and to upregulate miR-1 to inhibit aerobic glycolysis. Is there more insight into this conundrum? : This is a very difficult question to answer. microRNA (miR)-1 significantly abolishes the interaction between Smad3 and HIF-1α, which are attributed to reduction of HIF-1α, leading to suppress activation of Smad3 and reduce the expression of metabolic enzymes in the Warburg effect, such as HIF-1α, HK2 and MCT4, finally inhibits tumor proliferation. As shown in Figure 3, MCT4-expressing hypoxic cancer cells relatively depend on aerobic glycolysis, which is not consistent with the above findings. That is why I added the following sentence; “Surprisingly enough, miR-1 inhibits Warburg effect while Smad3 promotes it mediated by its downstream effecter HIF-1α and glycolytic enzymes such as HK2 and MCT4.”
Reviewer 3 Report
The review "The harmonious interplay of transporters induces the robustness of cancer cells" by Go J. Yoshida is a comprehensive overview of the role of amino acid and monocarboxylate transporters in cancer, related to metabolic heterogeneity-mediated acquisition of cancer cells survival advantage and resistance to therapy. The manuscript is well written, covering an important aspect of cancer metabolism, potentially interesting to a broad audience.
Author Response
Reviewer #3
The review "The harmonious interplay of transporters induces the robustness of cancer cells" by Go J. Yoshida is a comprehensive overview of the role of amino acid and monocarboxylate transporters in cancer, related to metabolic heterogeneity-mediated acquisition of cancer cells survival advantage and resistance to therapy. The manuscript is well written, covering an important aspect of cancer metabolism, potentially interesting to a broad audience. : I am deeply appreciated for your precious and kind comments. This review article mainly focuses on the role of amino acid and monocarboxylate transporters to maintain the phenotype associated with cancer stem cells. For example, xCT cystine transporter which is stabilized by CD44 variant at the cellular membrane renders cancer stem cells susceptible to sulfasalazine (SSZ), xCT inhibitor. As such, deep understanding of the function of transporters in malignancy enables us to discuss the novel therapeutic strategy.